# Predicting Tolerance to Cow’s Milk Allergy in Children Using IgE and IgG4 Peptide Binding Profiles

**DOI:** 10.3390/cells14050344

**Published:** 2025-02-27

**Authors:** Carlos Fernández-Lozano, Sergio Olmos-Piñero, Laura Sánchez-Ruano, Soledad Terrados, Mª del Carmen Diéguez, Montserrat Fernández-Rivas, Cristina Vlaicu, Inmaculada Cerecedo, Alejandro Gonzalo-Fernandez, Belén de la Hoz, Javier Martínez-Botas

**Affiliations:** 1Servicio de Bioquímica-Investigación, Hospital Universitario Ramón y Cajal, IRYCIS, 28034 Madrid, Spain; 2Servicio de Alergología, Hospital Universitario Ramón y Cajal, IRYCIS, 28034 Madrid, Spain; 3REI Network RD24/0007/0027, Instituto de Salud Carlos II, 28029 Madrid, Spain; mariamontserrat.fernandez@salud.madrid.org (M.F.-R.);; 4Servicio de Alergología, Hospital Clínico San Carlos, UCM, IdISSC, 28040 Madrid, Spain; 5ARADyAL Network RD16/0006/0009, Instituto de Salud Carlos III, 28029 Madrid, Spain; 6Servicio de Alergología, Hospital Universitario del Sureste, 28500 Madrid, Spain; 7Allergy Research Group, IdISSC, 28040 Madrid, Spain

**Keywords:** cow’s milk allergy, IgE, IgG4, peptide, biomarker, microarray, natural history, follow up, tolerance

## Abstract

Cow’s milk allergy (CMA) is the most common food allergy in infants. This study aimed to identify peptide biomarkers predictive of tolerance in a Spanish population of children with CMA. We investigated specific IgE and IgG4 binding to sequential epitopes of the five major CM allergens (α-s1-, α-s2-, β-, and κ-caseins as well as β-lactoglobulin) using a microarray-based immunoassay. Microarray analysis was performed in 118 patients at baseline and after 6, 18, 30, 42, and 54 months. Most patients tolerated CM at 6 months (40.7%) and 18 months (35.4%). We found significant differences in IgE and IgG4 binding intensity and diversity between allergic and tolerant patients. No differences were observed at baseline. Combining baseline IgE and IgG4 serology variables and peptide microarray analysis results, a predictive model was developed using the XGBoost algorithm to classify tolerance status at different time points. The generated models showed high predictive value at 6 and 30 months with AUCs of 0.883 and 0.833, respectively. Therefore, using IgE and IgG4 antibody-binding peptides at baseline, we generated two models predicting tolerance in children with cow’s milk allergy at 6 and 30 months.

## 1. Introduction

Cow’s milk (CM) is the first food introduced after birth or breastfeeding in western societies, so it is the first food antigen with which one comes into contact [1]. Cow’s milk allergy (CMA) is the most common food allergy in infants, affecting approximately 2.5% of children under 2 years of age [2]. CMA is usually a transient disease. Most children outgrow their allergy by 3 years of age, and only in a small percentage does it last longer than 4 years. These patients in particular often experience severe symptoms and lifelong persistence of the disease. The major allergens of CM are the coagulum proteins αS1-, αS2-, β-, and κ-casein as well as β-lactoglobulin (BLG) [3].

Peptide microarray-based analysis has become a powerful tool for identifying linear epitopes associated with food allergies in a high-throughput manner [4,5]. Previous work has demonstrated the importance of linear epitope recognition in the persistence and tolerance of cow’s milk allergy [6]. For instance, patients with persistent milk allergy have a higher linear epitope recognition than patients who have achieved tolerance [7,8,9,10,11,12]. Linear epitopes are also associated with the severity of allergic reactions against milk [11,12]. Furthermore, decreased IgE and increased IgG4 epitope binding has been associated with the achievement of tolerance in cow’s milk allergy [4,11]. Although IgE is crucial for allergenicity, IgG4 plays a central role in developing immune tolerance. Particularly, high levels of IgG4 antibodies to foods during infancy have been associated with earlier development of tolerance [13,14]. Also, maintaining tolerance to CM in atopic children and adults without CMA is associated with elevated levels of specific IgG4 (sIgG4) [15]. Although high levels of IgG4 antibodies have been associated with earlier development of tolerance, the role of IgG4 in allergic disease is controversial. It has been described that allergen-specific IgG4 plays a multifaceted role and may have a protective or pathogenic role depending on the allergens or exposure conditions [16].

Consequently, it is crucial to identify new biomarkers to be used by clinicians to predict who will develop tolerance more precisely. Our study aimed to identify novel peptide biomarkers to predict tolerance in cow’s milk allergy children. For this proposal, we analyzed the long-term changes in the IgE and IgG4 epitope binding profile of the five major CM proteins using peptide microarray.

## 2. Materials and Methods

### 2.1. Design and Setting

Patients included in this manuscript are part of the CoALE study, a longitudinal, prospective, multicenter, and coordinated study with a selection of incident cases. The selection was made prospectively and consecutively among the new patients referred to the three participating allergy services of Hospital Universitario Ramón y Cajal, Hospital Clínico San Carlos, and Hospital del Sureste, all in the Community of Madrid. All patients had an immediate reaction to cow’s milk, with positive SPT and cow’s milk specific IgE (>0.35 Ku /L). The patients who were selected had to meet the following inclusion criteria: (1) present immediate reactions (≤2 h) after the ingestion of cow’s milk or adapted formula and (2) show the first reaction within 6 months prior to the consultation. If patients did not have a reaction in the last 6 months, a double-blind placebo-controlled food challenge was performed. The project’s duration was 6 years, starting in 2010 and ending in 2016. For the patients included in this manuscript, serum samples were obtained to assess the IgE and IgG4 peptide recognition of allergenic proteins by peptide microarray.

The schedule of follow-up visits was as follows: visit 1 or baseline (0 months), visit 2 (6 months), visit 3 (18 months), visit 4 (30 months), visit 5 (42 months), and visit 6 (54 months). A positive oral challenge test established the clinical reactivity to CMA at each time of the study. An anamnesis was conducted for each patient, and clinical information was collected using a specific questionnaire [17].

All participants were tested for total IgE, specific IgE levels against whole milk, BLG, α-lactalbumin, and casein, and specific IgG4 against milk BLG, α-lactalbumin, and casein using the Phadia 250 ThermoScientific robot (ThermoFisher Scientific/Phadia, Uppsala, Sweden). DBPCFC tests were conducted following the procedure established in the SAFE [18] projects.

The ethics committee of the three participating centers approved the protocol on 25 May 2011: Hospital Universitario Ramón y Cajal, Hospital Clínico San Carlos, and Hospital del Sureste, B-08/174 (Ethical Approval Date: 25 May 2011). The parents or legal representatives of all participants signed an informed consent.

### 2.2. Peptide Microarray Immunoassay

The microarray printing was performed as previously described [19] with minor modifications. A library of overlapping peptides of 20 amino acids (AA) with an offset of 3 AA, corresponding to the primary sequences of α-s1 (bos d 9, 61 peptides), α-s2 (bos d 10, 64 peptides), β- (bos d 11, 64 peptides), and κ-caseins (bos d 12, 51 peptides), and BLG (bos d 5, 49 peptides), were commercially synthesized (GenScript Corporation, Piscataway, NJ, USA). Each peptide was printed twice. Between every two sets of peptides, 4 PBSs were printed as a negative control and for background normalization.

### 2.3. Data Analysis

Microarray analysis was performed according to the method of Lin et al. [20,21]. An individual peptide sample was considered positive if the standardized intensity of fluorescence (represented as weighted average Z-score) exceeds three. From these data, the following study variables were obtained: binding intensity (measured as average Z-score) and diversity of peptides (measured as the number of positive peptides with Z-score > 3). Data analysis and presentation were performed by using Microsoft Excel, and TIGR MultiExperiment Viewer (Mev v3.1) software. Cluster analysis was performed using the Pearson correlation average linkage method.

The informative peptides were defined as those recognized by more than 75% of allergic patients. The extreme gradient boosting (XGBoost) algorithm was combined with decision trees to develop the predictive model. This approach employs classification trees as weak learners, iteratively optimizing predictions by readjusting the residuals of previous models. In each iteration, greater weight is assigned to misclassified observations, improving model accuracy by adjusting the logistic loss function [22].

The variables analyzed to compile the bioinformatics models together with the peptides recognized by IgE and IgG4 were the baseline determinations of specific IgEs against cow’s milk, casein, beta-lactoglobulin, and alpha-lactalbumin and specific IgG4 against casein, beta-lactoglobulin, and alpha-lactalbumin. The dataset was divided into training (70%) and test (30%) subsets, using tenfold cross-validation to ensure model generalizability. Performance metrics included F1-score, precision, recall, and accuracy, calculated from confusion matrices. Additionally, ROC curves were generated to measure the area under the curve (AUC). Hyperparameter selection was conducted using a grid search approach combined with cross-validation, optimizing for the F1-score. The final number of trees and associated hyperparameters were determined based on the configuration that yielded the highest F1-score during the cross-validation process, ensuring a balance between model accuracy and the prevention of overfitting. All implementations were carried out with R software (version 4.3.1) using the caret (version 6.0) [23] and xgboost (version 1.7.7.1) libraries [22].

## 3. Results

### 3.1. Characteristics of the Patients

The CoALE cohort recruited 152 consecutive patients diagnosed with CMA. In the present study, we included the 118 patients in whom peptide microarrays tests were performed. The clinical characteristics of the patients are shown in Table 1. At baseline the highest value of serum IgE was obtained in whole milk with a median of 0.86 kU/L (IQR 0.38–2.64), while the values of α-lac, β-lac, or casein were below 0.40 kU/L. The median of the total IgE levels in the basal visit was 14.70 kU/L (IQR 7.80–30.50).

Among the 118 patients, the percentages of tolerant patients at each visit were as follows: 40.7% of patients were tolerant at visit 2, 35.4% at visit 3, 22.2% at visit 4, 22.2% at visit 5, and 7.1% at visit 6. Most patients tolerated CM at 6 and 18 months from the start of the study, which corresponds to an age of 1.5–2 years. At the end of the study, 13 patients remained allergic, 76 were tolerant, and 29 voluntarily withdrew from the study (lost patients) (Figure 1).

### 3.2. IgE and IgG4 Antibody Binding to Sequential Epitopes Reveled Four Informative Epitopes Recognized in All-Time Points

We examined the IgE and IgG4 binding to sequential epitopes of the five major CM allergens using a peptide microarray corresponding to linear sequences of α-s1-, α-s2-, β-, and κ-caseins as well as BLG. The microarray analysis was performed on 118 patients at different time points until they could tolerate the food, at which point they left the study.

Figure 2 it shows the IgE recognition pattern and TileMap representation of the informative epitopes. We considered an IgE-binding peptide as an informative epitope if they were recognized by more than 75% of allergic patients, as described in the Section 2. We found eight informative epitopes at baseline, six at visit 2 (T6), 15 at visit 3 (T18), 14 at visit 4 (T30), 32 at visit 5, and 39 at visit 6 (Figure 2). Four informative epitopes were recognized at all-time points (αs1-cas 3, αs1-cas 16–18, κ-cas 29–31, and β-lac 37–39).

In addition, we found significant differences in the IgE-binding intensity (measure as average Z-score) and diversity (measure as the number of positive peptides with Z-score > 3). At visits 2 (T6), 3 (T18), and 4 (T30), allergic patients recognized significantly more peptides and more intense than tolerant individuals of αs1-cas, β-cas, and κ -cas (Appendix A). It is also important to mention that no differences were found at baseline, either in IgE-binding intensity or in the number of peptides. For patients’ classification at the baseline visit, where all patients were allergic, the 6-month tolerance status was used (visit 2) (Appendix A).

The IgG4 binding was relatively weaker compared with the IgE in all milk peptides, particularly at baseline (Figure 3). Only one informative epitope was found at visit 5 (T48) (β-lac 41–43). Similarly to IgE, we have also found differences in the IgG4-binding intensity and diversity for αs1-cas, β-cas, and κ -cas at visits 2 (T6), 3 (T18), and 4 (T30) (Appendix A). Like IgE, there were no significant differences at the baseline visit (Appendix A).

### 3.3. Combining Serological Variables and Peptide Biomarkers at Baseline Can Predict Tolerability at Later Time Points

Next, we test the utility of the serology variables, including milk sIgE, α-lactalbumin sIgE, BLG sIgE, casein sIgE, total IgE, and IgE and IgG4 antibody binding to sequential epitopes, to predict tolerance in children with CMA. These serology variables and the epitope binding at baseline (visit 1-T0) were combined into a single model and classified using the extreme gradient boosting (XGBoost) algorithm to determine the utility to predict the tolerance at visit 2 (T6), visit 3 (T18), and visit 4 (T30). The analyses of visits 5 (T42) and 6 (T42) were not performed due to the reduced number of remaining allergic patients at these time points, 14 and 13, respectively. Next, ROC curves were generated to evaluate their diagnostic performance in both training and validation groups. The general performance of the models was evaluated using several metrics (AUC, sensitivity, specificity, F1 score, and recall) and presented as radar charts.

As can be seen in Figure 4, the model generated to predict tolerance at visit 2 (T6) has a good diagnostic performance with an AUC of 0.908 in the training group and 0.883 in the testing group (Figure 4A) and excellent metrics (Figure 4B). Milk sIgE is the predominant variable, and the rest of the variables contributed to a lesser extent (milk sIgG; IgE αs1-cas p1, p4, and p18; IgE β-cas p53 and 54; and IgE κ-cas p33) (Figure 4C).

In relation to the predictive model at visit 3 (T18), although the ROC curve of the training group has a good AUC of 0.89, the AUC of the testing group is reduced to 0.66 (Figure 5A) and would not be a good predictive model according to the metrics (Figure 5B). In this case, as in the previous model, the predominant variable would be the milk sIgE, followed by the IgE κ-cas p50, α-lactalbumin sIgE, and to a lesser extent lesser extent milk sIgG, IgE β-lac p37, casein sIgE, IgE β-cas p26, β-lactoglobulin sIgE, IgE β-lac p25, and IgE β-lac p 46 (Figure 5C).

Finally, the model generated to predict tolerance at visit 4 (T30) has a good diagnostic performance with AUC values of 0.993 in the training group and 0.833 in the testing groups (Figure 6A). The associated metrics confirm the goodness of fit of the model (Figure 6B). In this case, the variables distribute their importance in a much more balanced way giving greater importance to the information provided by the peptides in the classification (Figure 6C). The main variables are α-lactalbumin sIgE, IgE αs1-cas p18, BLG sIgE, casein sIgE, IgE αs1-cas p2, milk sIgE, IgG4 β-lac p24, IgE κ-cas p29, IgE κ-cas p51, and β-cas p52.

Therefore, using serological variables IgE and IgG4 against milk and its proteins and IgE and IgG4 antibody binding to sequential epitopes at baseline, we generated two models that may be useful in predicting tolerance in children with CMA at later time points of 6 and 30 months.

## 4. Discussion

In the present work, we studied 118 consecutive patients diagnosed with CMA and analyzed the IgE and IgG4 binding profile to sequential epitopes of the five major CM allergens using a peptide microarray corresponding to linear sequences of α-s1-, α-s2-, β-, and κ-caseins as well as BLG. To our knowledge, this is the first study that analyzed the B-cell peptides using a longitudinal and prospective design that covered the natural history of CMA from the diagnosis to 54 months of follow-up that includes consecutively recruited patients. A very homogeneous sample was obtained with a rigorous study that included the provocation test as a method to identify tolerant and allergic patients. In our population, 40% of patients tolerated the milk after 6 months of follow-up. Similar to other cohorts, the tolerance percentage of patients decreases with age. This fact is related to natural tolerance development. Thus, in our study, most children outgrow their allergy by 3 years of age, and only a small percentage persisting beyond 4 years [24].

In the present work, we identified four key epitopes (αs1-cas 3, αs1-cas 16–18, κ-cas 29–31, and β-lac 37–39) recognized by more than 75% of milk-allergic patients over time, suggesting a link to allergy persistence. In addition, we developed two predictive models of tolerance at 6 and 30 months, using the baseline IgE and IgG4 antibody-binding peptides in conjunction with standard serological variables. These biomarker peptides may be useful in the development of new diagnostic tests, peptide-based synthetic vaccines, or hypoallergenic milk formulas.

According to our results, the role of IgG4 epitopes plays a minor role in the development of tolerance because the IgG4 binding was relatively weaker compared to IgE for all milk peptides, and only one informative epitope was found at visit 5 (compare Figure 2 and Figure 3). On the other hand, only one IgG4 peptide was selected among the highest weights in all three models (IgG4 β-lac p 24, Figure 6).

CMA is a dynamic disease in which patients acquire spontaneous tolerance in a high percentage. Although the diagnosis of CMA is usually made by skin testing and the assessment of IgE antibodies to whole milk or specific milk proteins, in many cases, this does not preclude the use of challenge testing for follow-up or determining when the patient has lost clinical reactivity [25].

Two previous studies have addressed the development of natural tolerance using microarray epitope mapping. Savilahti et al. [11] analyzed the epitope mapping at different time points (at diagnosis, one year after diagnosis, and at follow-ups of 8 to 9 years), comparing the profile between patients with persisting IgE-mediated CMA at age 8 to 9 years and patients who recovered by age 3 years. They found that IgE and IgG4 binding to a panel of 15 CM peptides at the time of diagnosis predicted with significant accuracy whether a child would recover from CMA early or have a persisting allergy.

Caubet et al. performed a retrospective study that combined the data of 53 time points with proven reactivity to CM (positive DBPCFC) from 35 children with CMA and compared those with data of 19 time points when tolerance has developed (median age at tolerance acquisition of 51 months). They identified several epitopes with significantly greater IgE binding from patients with persistent allergy beyond 5 years of age compared with those with early recovery (outgrown before 5 years) [26].

In contrast to these studies, in the present work, we analyzed the tolerance status and its correlation with the epitope recognition profile at each time point (0, 6, 18, and 30 months). Although the studies are not comparable due to their different designs, it is interesting that we were able to associate specific linear epitope profiles with tolerance status in all three studies, which underlines the importance of linear epitopes as biomarkers to predict tolerance.

In agreement with these studies, we also observed no differences in IgE and IgG4 intensity and the diversity of peptide recognition between patients at the time of diagnosis. We compared the basal IgE and IgG4-binding patterns in our case and classified the patients according to the six-month tolerance status. Therefore, it seems that all patients are similar at the time of diagnosis, and it is later, after six months, when differences begin to be observed, coinciding with the maximum peak of tolerance.

In our study, we have established models of peptide recognition at time zero that differentiate tolerance to milk at subsequent study times, complementing the determination of specific IgE to CM, which is particularly relevant in predicting tolerance at 6 and 30 months. However, the longer the patients remain allergic, the more useful the baseline detection pattern becomes in predicting tolerance in older patients. This is evident in the prediction algorithm at 30 months. This may be due to the selection of the most persistent patient profiles, which would allow us to identify patients with a poorer prognosis at diagnosis from the onset of the disease. The role of casein-specific IgE in disease persistence has been described in previous studies, with high levels maintained over time being associated with a lower chance of tolerance [27]. In this context, it is interesting that in the 30-month follow-up in our study, the model included casein-specific IgE, and it appears as a component of the model above for cow’s milk IgE.

Our findings allow us to identify patients at higher risk of not achieving spontaneous tolerance and, therefore, identify patients who could benefit from early intervention. Patients could be identified as candidates for immunotherapy against cow’s milk due to their risk of developing persistent allergies in the future. In addition, the identification of linear peptides plays a role in the follow-up of milk immunotherapy in allergic patients, as previously demonstrated by our group [19]. The usefulness of peptides in the follow-up of oral immunotherapy with other foods has also been found for eggs and peanut [28,29]. Knowledge of B-reactive epitopes throughout the patients’ natural history would also allow for the design of more selective vaccines.

Several studies have sought to establish cut-off points for skin test sizes and specific IgE levels in order to assess the likelihood of tolerance during a challenge test. However, these thresholds are not universally accepted, as their accuracy can vary based on factors such as the prevalence of the condition within a given population. Even among similar groups of patients, some individuals may tolerate milk, while others do not [4,30]. In this context, it is interesting that our results were analyzed each time by comparing the profile of peptide recognition between tolerant and allergic patients. We found significant differences in the intensity and diversity of IgE and IgG4 binding between allergic and tolerant patients at all study times. At baseline, no differences were found in IgE and IgG4 binding intensity or number of peptides. Other authors have analyzed the recognition profile between tolerant and patients who remained allergic. Cerecedo et al. compared the IgE/IgG4 ratio of epitope binding between reactive and tolerant patients and found several regions associated with clinical reactivity to milk [4]. Later, Matsumoto et al. found that the distribution of IgE and IgG4 could differentiate between persistent CMA patients and those who had outgrown CMA or sensitized patients without CMA [10]. They suggested that changes in the IgE/IgG4 ratio could be used to estimate the likelihood of outgrowing CMA. Consequently, these studies, including the present study, suggest that the determination of peptide recognition profiles, together with the classical clinical variables, could be a very useful tool for the indication of a cow’s milk food challenge with a higher chance of success.

Peptide microarrays are innovative diagnostic tools that allow the determination of the diversity and intensity of IgE and IgG4 antibody binding to sequential epitopes [31]. These arrays allow the quantification of IgE and IgG levels to cow’s milk (CM) allergens, facilitating their classification into relevant and non-relevant epitopes [32]. Such tests are essential for studying the early development of allergic immune responses in early childhood and for monitoring immune responses during natural tolerance induction [33]. This is the first study that identifies linear epitopes with prognostic capacity and establishes their validity as a diagnostic test against oral challenge through the follow-up of the natural history of the disease. This gives us an essential tool to improve diagnosis. On the one hand, they allow us to identify the diversity of IgE antibodies binding to sequential epitopes and give us a decision value for the oral challenge that can be used to decide the timing of the food challenge test. We propose two predictive models based on peptide recognition and common serological variables at the time of diagnosis that may be useful in predicting the subsequent development of CM tolerance. If the linear B epitopes identified in our study are validated as indicators of CM tolerance or persistence, their determination will allow us to have personalized and precise therapeutic management from the onset of CMA in the future.

## 5. Conclusions

In the present work, we identified four key epitopes (αs1-cas 3, αs1-cas 16–18, κ-cas 29–31, and β-lac 37–39) recognized by more than 75% of milk-allergic patients over time, suggesting a link to allergy persistence. These epitopes may be good candidates for the development of peptide vaccines or hypoallergenic milk formulas. In addition, we developed two predictive models of tolerance at 6 and 30 months, using the baseline IgE and IgG4 antibody-binding peptides in conjunction with standard serological variables. Further investigation is required to validate the use of these peptide biomarkers.

## Figures and Tables

**Figure 1 cells-14-00344-f001:**
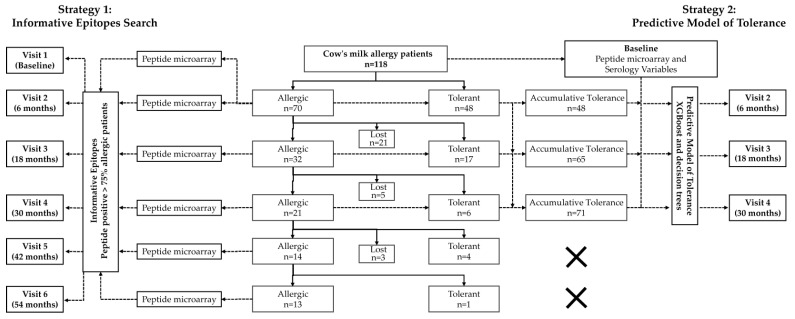
Study design and analysis strategy. Strategy 1 for the search for informative epitopes: the informative peptides were defined as those recognized by more than 75% of allergic patients at each time point. Strategy 2 for a predictive model of tolerance: the XGBoost algorithm was combined with decision trees to develop the predictive model of tolerance at visit 2 (T6), visit 3 (T18), and visit 4 (T30). Each model combined the serological variables and epitope binding at baseline (visit 1-T0) with the tolerance status (allergic or tolerant) at each time point. Patients were defined as tolerant if tolerated at or before the specified time (accumulative tolerance). The analyses of visits 5 (T42) and 6 (T42) were not performed due to the reduced number of remaining allergic patients. Lost patients voluntarily withdrew from the study.

**Figure 2 cells-14-00344-f002:**
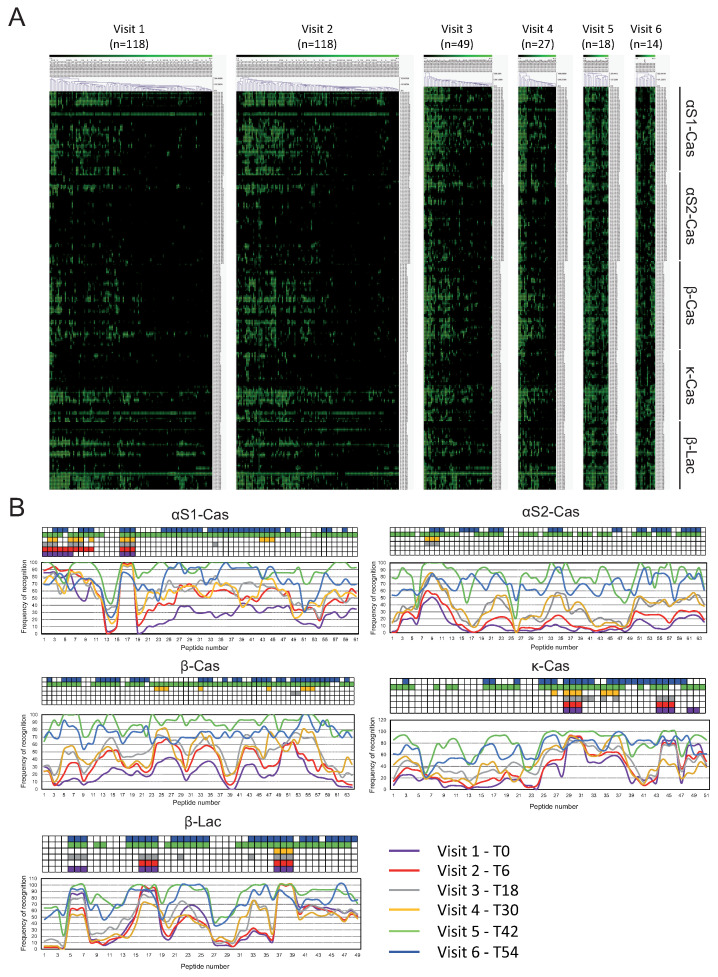
(**A**) IgE recognition pattern of peptides of αs1-cas, αs2-cas, β-cas, κ-cas, and β-lac proteins after grouping by hierarchical clustering at visits 1 (0 months), 2 (6 months), 3 (18 months), 4 (30 months), 5 (42 months), and 6 (54 months). The intensity of IgE binding is represented by weighted average Z-scores on a grading scale according to the scale bar at the top of the figure. (**B**) Percentage of IgE recognition of peptides of αs1-cas, αs2-cas, β-cas, κ-cas, and β-lac proteins. The y-axis shows the percentage of patients within each visit showing positive binding (Z-score < 3) to each peptide. The x-axis shows the number of peptides for each protein. The TileMap on top of each protein represents the informative epitopes that were recognized by more than 75% of allergic patients within each visit according to the color code of each visit.

**Figure 3 cells-14-00344-f003:**
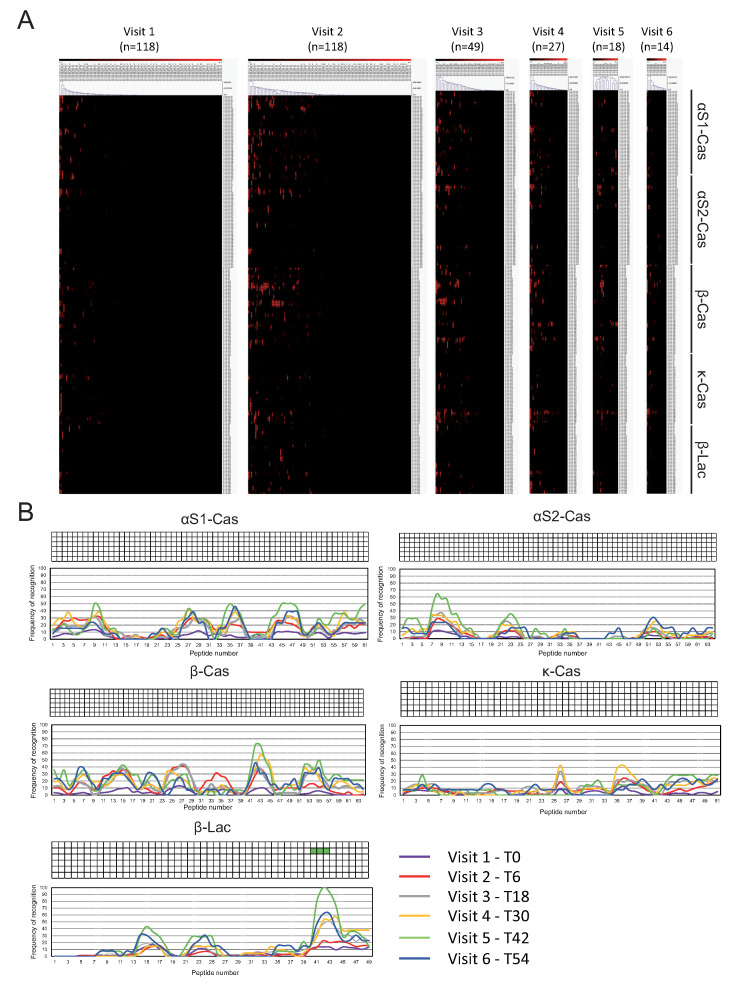
(**A**) IgG4 recognition pattern of peptides of αs1-cas, αs2-cas, β-cas, κ-cas, and β-lac proteins after grouping by hierarchical clustering at visits 1 (0 months), 2 (6 months), 3 (18 months), 4 (30 months), 5 (42 months), and 6 (54 months). The intensity of IgG4 binding is represented by weighted average Z-scores on a grading scale according to the scale bar at the top of the figure. (**B**) Percentage of IgG4 recognition of peptides of αs1-cas, αs2-cas, β-cas, κ-cas, and β-lac proteins. The y-axis shows the percentage of patients within each visit showing positive binding (Z-score < 3) to each peptide. The x-axis shows the number of peptides for each protein. The TileMap on top of each protein represents the informative epitopes that were recognized by more than 75% of allergic patients within each visit according to the color code of each visit.

**Figure 4 cells-14-00344-f004:**
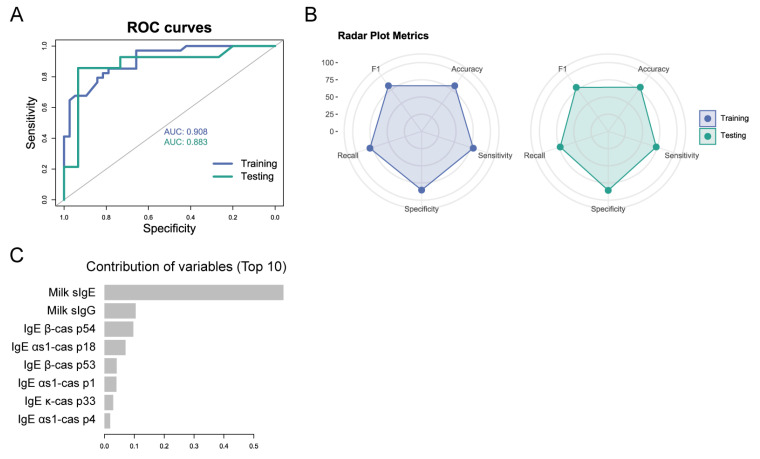
Metrics of the model to predict tolerance at visit 2 (T6). (**A**) ROC curve of the training and the testing group. (**B**) Radar plot of the metrics of the training and the testing group. (**C**) Contribution of the top 10 variables to the predicting model.

**Figure 5 cells-14-00344-f005:**
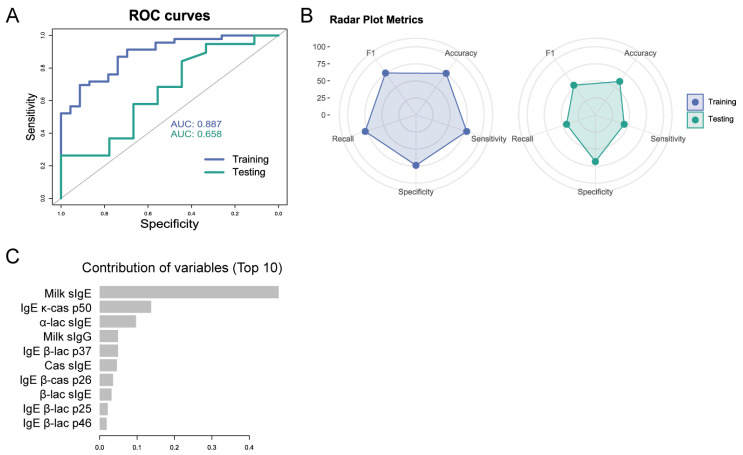
Metrics of the model to predict tolerance at visit 3 (T18). (**A**) ROC curve of the training and the testing group. (**B**) Radar plot of the metrics of the training and the testing group. (**C**) Contribution of the top 10 variables to the predicting model.

**Figure 6 cells-14-00344-f006:**
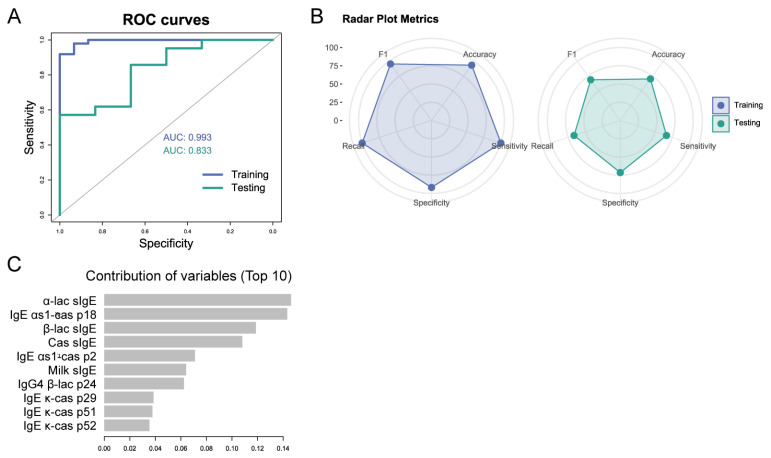
Metrics of the model to predict tolerance at visit 4 (T30). (**A**) ROC curve of the training and the testing group. (**B**) Radar plot of the metrics of the training and the testing group. (**C**) Contribution of the top 10 variables to the predicting model.

**Table 1 cells-14-00344-t001:** Basal characteristics of patients in the CoALE study.

Characteristics	Number
Total number of patients	118
Age at initial visit, median (IQR)	6.2 mo. (4.8–7.44)
Sex, n (%)	
Female	58 (49.2%)
Male	60 (50.8%)
Ethnicity, n (%)	
Caucasians	107 (90.68%)
Hispanics	9 (7.62%)
Others	2 (1.68%)
Atopic dermatitis, n (%)	52 (44.44%)
Breastfeeding, n (%)	118 (100%)
Neonatal bottle, n (%)	55 (46.61%)
Initial symptoms of CMA, n (%)	
Oral	29 (24.58%)
Cutaneous	102 (86.44%)
Gastrointestinal	54 (45.76%)
Respiratory	9 (7.63%)
Cardiovascular	2 (1.69%)
Neurological	1 (0.85%)
Total IgE, kU/L, median (IQR)	14.35 (7.88–28.03)
Specific IgE, kU/L, median (IQR)	
Milk	0.89 (0.35–2.64)
α-Lactalbumin	0.35 (0.17–0.80)
β-Lactoglobulin	0.39 (0.30–1.26)
Casein	0.34 (0.13–0.72)
Specific IgG4, mg/L, median (IQR)	
α-Lactalbumin	0.08 (0.01–0.35)
β-Lactoglobulin	0.16 (0.07–0.82)
Casein	0.11 (0.05–0.26)

## Data Availability

The raw data supporting the conclusions of this article will be made available by the authors, without undue reservation.

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
