# Peer review of "Predicting Tolerance to Cow’s Milk Allergy in Children Using IgE and IgG4 Peptide Binding Profiles"

_cells, 2025, doi:10.3390/cells14050344_

Round 1

Reviewer 1 Report

Comments and Suggestions for Authors

This is an important topic, mostly because unlike other food allergens, the ability to undergo oral immunotherapy is limited in a proportion of milk allergic patients. It is important to identify patients with a higher risk to stay allergic and maybe even detect patients who will show difficulty to achieve tolerance when undergoing oral immunotherapy. However, I found it hard to follow the methodology of this study and this is what I address in most of my comments.

Major comments:

1.      Study design-

It is not clear to me how patients were chosen and how the CM allergy was defined. If oral food challenges were conducted, what was the threshold?

2.      Study flow-

The number of patients is not clear enough throughout the manuscript. It is not clear whether all the 118 are allergic (from the supplementary material I learned they were not). It is also not clear why the numbers changed from visit to visit- where patients lost to follow up? The decrease in tolerance is not clear either.

3.      Discussion-

The discussion is quite long and the major findings and their importance are not described clear enough.  

Minor comments:

Introduction-

Lines 53-54- In my opinion "the status of tolerance" can be defined by an oral food challenge. Epitope binding can help predict who will develop tolerance, I do not think it can define the status of tolerance.

Methods-

Are addressed to the Design and setting section:

Important clinical and demographic information is missing:

Patient age at diagnosis, other atopic background, the initial reaction to CM, how were they defined allergic? If by OFC, then what was the median mg of protein that caused a reaction.

If this is described elsewhere it should at least be added as supplemental material.

It is very important to clarify how CM Allergy was diagnosed. Please clarify whether a history of reaction plus SPT or sIgE, or whether an OFC was performed.

All the information depicted in Lines 69-77 is not described in this study. Not everything is relevant, such as FAQLQ, but all clinical and demographic data is.

Results-

Table 1 does not really include clinical data, or rather much data is missing.

According to remark above, data should be given regarding patient's history of CMA

Only in the supplemental material we learn that 70 were allergic and 48 tolerant at baseline. This should be mentioned in the results section. Also, because these patients are compared, they should be compared in Table 1 as well.

Lines 127-129- should be moved to the Methods section to describe the definition of CMA. However, it is important to understand to what amount of protein there was a reaction, since this can maybe correlate to the epitope mapping.

Lines 130-133 are not clear to me- How come less and less patients were tolerant from visit to visit? As described in literature and in references given by the authors, children achieve more tolerance as they grow older

Line 141- the number of patients described in figures 1 and 2 are the ones who did NOT develop tolerance? Again, please clarify

Lines 151-156- the results of Figure S1 should be described before figure S2-4.

Lines 159-162- S5 should be described before S6-8

Figure S2- has 3 asterisks, please define

Line 188- please correct "test" to "tested"

Discussion

Line 238- "118 patients diagnosed with CMA". Earlier we learned that only 70 were diagnosed. Again, the numbers are confusing.

Lines 245-247- Again, I do not understand why the tolerance of patients decreases with time. The reference given (23) also depicts increased tolerance over time.

Lines 270- 275- this paragraph aims to describe the main finding of the study. I think it is important to describe them in a more accurate and precise way, since they are not clear enough to the reader. These should also be described shortly in the first paragraph.

The discussion is missing the main implementations of these findings into clinical scenarios.

Conclusion

This should summarize precisely and shortly the main findings and their importance. Instead of "we found significant differences…" or "allowed us to identify peptides…" it should describe exactly what was found and it's importance.    

Author Response

Dear reviewer,

Thank you for your careful review of our paper and for the valuable comments, corrections, and suggestions that have helped us to improve the manuscript. Below, you will find the point-by-point response to your questions.

Major comments:

  1. Study design-

It is not clear to me how patients were chosen and how the CM allergy was defined. If oral food challenges were conducted, what was the threshold?

The selection was made prospectively and consecutively among the new patients referred to the three participating allergy services of Hospital Universitario Ramón y Cajal, Hospital Clínico San Carlos, and Hospital del Sureste, all in the Community of Madrid. All patients had an immediate reaction to cow's milk, with positive SPT and cow's milk specific IgE. See line 76.

Participants must meet the following inclusion criteria: (1) show immediate reactions (≤2 hours) after ingestion of cow's milk or adapted formula and (2) show the first reaction within 6 months prior to the visit. At the first visit, most patients had a recent reaction and no provocation was performed according to the protocol. At subsequent visits, the vast majority were provoked because they were on a non-dairy diet and did not report a recent reaction.

The threshold dose for tolerance was 200 mL.

To better clarify this point we added the following sentences to Methods:

“All patients had an immediate reaction to cow's milk, with positive SPT and cow's milk specific IgE (>0.35 Ku /L). The patients who were selected had to meet the following inclusion criteria: (1) present immediate reactions (≤2 hours) after the ingestion of cow's milk or adapted formula and (2) show the first reaction within 6 months prior to the consultation. If patients did not have a reaction in the last six months, a double-blind placebo-controlled food challenge was performed.”. Page 2, line 76.

  1. Study flow-

The number of patients is not clear enough throughout the manuscript. It is not clear whether all the 118 are allergic (from the supplementary material I learned they were not). It is also not clear why the numbers changed from visit to visit- where patients lost to follow up? The decrease in tolerance is not clear either.

The number of patients changes as tolerant patients finish the study by protocol.  There are also patients who voluntarily leave the study (lost patients). To clarify this point, we have added a new supplementary figure showing the study design and clearly indicating which patients are tolerant, allergic or leaving the study at each visit. (Figure S1).

  1. Discussion-

The discussion is quite long and the major findings and their importance are not described clear enough.  

To better describe the major finding and their importance we add the following paragraph to the Discussion section:

“In the present work, we identified four key epitopes recognized by more than seventy-five percent of milk-allergic patients over time, suggesting a link to allergy persistence. In addition, we developed two predictive models of tolerance at 6 and 30 months, using the baseline IgE and IgG4 antibody-binding peptides in conjunction with standard serological variables. These biomarker peptides may be useful in the development of new diagnostic tests, peptide-based synthetic vaccines, or hypoallergenic milk formulas.” Page 9, line 269.

Minor comments:

Introduction-

Lines 53-54- In my opinion "the status of tolerance" can be defined by an oral food challenge. Epitope binding can help predict who will develop tolerance, I do not think it can define the status of tolerance.

Following your recommendation we replace "define the status of tolerance" by “predict who will develop tolerance” in that sentence. Page 2, line 64.

Methods-

Are addressed to the Design and setting section:

Important clinical and demographic information is missing:

Patient age at diagnosis, other atopic background, the initial reaction to CM, how were they defined allergic? If by OFC, then what was the median mg of protein that caused a reaction.

If this is described elsewhere it should at least be added as supplemental material.

It is very important to clarify how CM Allergy was diagnosed. Please clarify whether a history of reaction plus SPT or sIgE, or whether an OFC was performed.

All the information depicted in Lines 69-77 is not described in this study. Not everything is relevant, such as FAQLQ, but all clinical and demographic data is.

Following your recommendation, we add a new supplementary table with the clinical and demographic information (Table S1).

To clarify these points, we have added the following sentences to the Design and Setup section:

“All patients had an immediate reaction to cow's milk, with positive SPT and cow's milk specific IgE (>0.35 Ku /L). The patients who were selected had to meet the following inclusion criteria: (1) present immediate reactions (≤2 hours) after the ingestion of cow's milk or adapted formula and (2) show the first reaction within 6 months prior to the consultation. If patients did not have a reaction in the last six months, a double-blind placebo-controlled food challenge was performed.” Page 2, line 76.

“A positive oral challenge test established the clinical reactivity to CMA at each time of the study.” Page 2, line 87.

Results-

Table 1 does not really include clinical data, or rather much data is missing.

According to remark above, data should be given regarding patient's history of CMA

We add a new supplementary table with the clinical and demographic information (Table S1).

Only in the supplemental material we learn that 70 were allergic and 48 tolerant at baseline. This should be mentioned in the results section. Also, because these patients are compared, they should be compared in Table 1 as well.

To clarify this point, we have added a new supplementary figure showing the study design and clearly indicating which patients are tolerant, allergic or withdrawing from the study at each visit. (Figure S1).

All patients were allergic at baseline and we used the biochemical and peptide binding parameters at that time to predict who would tolerate and who would not at later times (6, 18 and 30 months).

En the other hand, due to the fact that all patients were allergic at baseline, we only used the six-month qualification for this time to compare the intensity or diversity of peptide recognition at baseline (Figure S5 and S9). We use the tolerance to its own time for the other comparisons (Figure S2, S3, S4, S6, S7, and S8).

Lines 127-129- should be moved to the Methods section to describe the definition of CMA. However, it is important to understand to what amount of protein there was a reaction, since this can maybe correlate to the epitope mapping.

Following your recommendation, we move these sentences to the method section:

“The schedule of follow-up visits was: visit 1 or baseline (0 months), visit 2 (6 months), visit 3 (18 months), visit 4 (30 months), visit 5 (42 months), and visit 6 (54 months). A positive oral challenge test established the clinical reactivity to CMA at each time of the study. An anamnesis was conducted for each patient, and clinical information was collected using a specific questionnaire including FAQLQ-PF Quality of Life Questionnaire [15].” Page 2, Line 86.

Regarding the question of reaction dose, this is a very interesting point, but it cannot be answered by the design of our study, because at the first visit most of the patients had a recent reaction and according to the protocol, if the reaction had occurred within the last 6 months, no provocation was performed. It is important to note that at subsequent visits, the vast majority were provoked because they were on a milk-free diet and did not report a recent reaction.

Lines 130-133 are not clear to me- How come less and less patients were tolerant from visit to visit? As described in literature and in references given by the authors, children achieve more tolerance as they grow older

It is not that less and less patients were tolerant from visit to visit, but rather that, according to the protocol, tolerant patients leave the study and do not return for subsequent visits. At the end of the study, 13 patients remained allergic, 76 were tolerant and 29 voluntarily withdrew from the study. Therefore, we use the accumulated tolerant patients for the identification of biomarker peptides. To clarify this point, we have added a new supplementary figure showing the study design and clearly indicating which patients are tolerant, allergic or withdrawing from the study at each visit. (Figure S1).

Line 141- the number of patients described in figures 1 and 2 are the ones who did NOT develop tolerance? Again, please clarify

All patients were allergic at baseline (visit 1) (n=118), all allergic patients returned at visit 2 (n=118), and after oral challenge some patients tolerated (n=48) and others remained allergic (n=70). Tolerant patients left the study and did not return for visit 3. All allergic patients were expected to return for visit 3, but 21 left the study voluntarily and only 49 returned. Similarly, there were 27 patients at visit 4, 18 at visit 5 and 14 at the final visit. The new Supplementary Figure 1 summarizes all this information.

Lines 151-156- the results of Figure S1 should be described before figure S2-4.

We have rearranged the figures to match how they appear in the text.

Lines 159-162- S5 should be described before S6-8

We have rearranged the figures to match how they appear in the text.

Figure S2- has 3 asterisks, please define

We have added the 3 asterisks value to the figure legs.

Line 188- please correct "test" to "tested"

We have amended it. 

Discussion

Line 238- "118 patients diagnosed with CMA". Earlier we learned that only 70 were diagnosed. Again, the numbers are confusing.

To clarify this point, we have added a new supplementary figure showing the study design and clearly indicating which patients are tolerant, allergic or withdrawing from the study at each visit. (Figure S1).

Lines 245-247- Again, I do not understand why the tolerance of patients decreases with time. The reference given (23) also depicts increased tolerance over time.

See the comment above.

Lines 270- 275- this paragraph aims to describe the main finding of the study. I think it is important to describe them in a more accurate and precise way, since they are not clear enough to the reader. These should also be described shortly in the first paragraph.

Unfortunately, we did not express ourselves very well in this paragraph and what we wanted to say was that, despite the different design of the three studies, in all of them we were able to associate the recognition of linear peptides with the development of tolerance.

We have made the following change to the paragraph to improve the message:

“Although the studies are not comparable due to their different designs, it is interesting that we were able to associate specific linear epitope profiles with tolerance status in all three studies, which underlines the importance of linear epitopes as biomarkers to predict tolerance.” Page 10, line 300.

Following you recommendation we have also improve the description of the main finding of the study in the first paragraph:

“In the present work, we identified four key epitopes recognized by more than seventy-five percent of milk-allergic patients over time, suggesting a link to allergy persistence. In addition, we developed two predictive models of tolerance at 6 and 30 months, using the baseline IgE and IgG4 antibody-binding peptides in conjunction with standard serological variables. These biomarker peptides may be useful in the development of new diagnostic tests, peptide-based synthetic vaccines, or hypoallergenic milk formulas.” Page 9, line 269.

The discussion is missing the main implementations of these findings into clinical scenarios.

Following your recommendations, we have added the following paragraph to the discussion:

“If the linear B epitopes identified in our study are validated as indicators of CM tolerance or persistence, their determination will allow us, in the future to have personalized and precise therapeutic management from the onset of CMA. disease. If the linear B epitopes identified in our study are validated as indicators of persistence, their determination will allow us to provide personalized and precise therapeutic management from the onset of CMA.” Page 11, line 365.

Conclusion

This should summarize precisely and shortly the main findings and their importance. Instead of "we found significant differences…" or "allowed us to identify peptides…" it should describe exactly what was found and it's importance.    

Following your recommendations, we have modified the conclusion as follows:

“In the present work, we identified four key epitopes recognized by more than seventy-five percent of milk allergic patients over time, suggesting a link to allergy persistence. These epitopes may be good candidates for the development of peptide vaccines or hypoallergenic milk formulas. In addition, we developed two predictive models of tolerance at 6 and 30 months, using the baseline IgE and IgG4 antibody-binding peptides in conjunction with standard serological variables.” Page 11, line 372.

Reviewer 2 Report

Comments and Suggestions for Authors

This paper analyzes the binding of specific IgE and IgG recognizing sequential epitopes of the five major milk allergens and their diversity, reporting the utility of these markers in predicting the acquisition of tolerance to milk allergies. The results are intriguing in understanding the natural course of tolerance acquisition and changes in the recognized epitopes. The value of this paper would be enhanced by adding considerations on several points.

Specific Comments

1. The authors have created prediction models. ,How each values of epitope specific IgE and IgG4 were used in the predictive models should be specifically indicated.

2. Discussion on the clinical significance of the diversity of IgG4-specific epitope recognition and its association with tolerance acquisition is lacking. When comparing IgE-binding epitopes and IgG4-binding epitopes, does the data suggest that IgG4 inhibits the binding of IgE to antigens? Please discuss the role of epitope specific IgG4 in the blocking effects during tolerance acquisition.

3. When comparing Figures 3c, 4c, and 5c, the epitopes that are considered to have a significant impact in the prediction model differ across observation periods. What is the clinical significance of the change in important epitopes over time during the tolerance acquisition? Is this merely a statistical error, or does it imply that the epitopes gradually shift to adjacent regions? It should be discussed in light of the positional relationship of the epitope in the allergen molecules

Author Response

Dear reviewer,

Thank you for your careful review of our paper and for the valuable comments, corrections, and suggestions that have helped us to improve the manuscript. Below, you will find the point-by-point response to your questions.

Specific Comments

  1. The authors have created prediction models. How each values of epitope specific IgE and IgG4 were used in the predictive models should be specifically indicated.

The XGBoost algorithm uses peptides as predictor variables and we do not have the ability to set explicit classification thresholds. To better understand how it works and to try to identify the impact of each epitope on the decision, we performed a feature importance analysis (Figures 3, 4 and 5), which allows us to identify epitopes with a higher weight in the decision. In the future, using the epitopes with the greatest weight in the prediction, we will be able to combine them in a single device and validate it in another population, at which point thresholds for each epitope can be proposed.

  1. Discussion on the clinical significance of the diversity of IgG4-specific epitope recognition and its association with tolerance acquisition is lacking. When comparing IgE-binding epitopes and IgG4-binding epitopes, does the data suggest that IgG4 inhibits the binding of IgE to antigens? Please discuss the role of epitope specific IgG4 in the blocking effects during tolerance acquisition.

According to our results, the role of IgG4 epitopes plays a minor role in the development of tolerance, because the IgG4 binding was relatively weaker compared to IgE for all milk peptides and only one informative epitope was found at visit 5 (compare Figures 1 and 2). On the other hand, only one IgG4 peptide was selected among the highest weights in all three models (IgG4 β-lac p 24, Figure 5).

Following your recommendation, we introduced this paragraph in the Discussion. Page 9, line 275.

  1. When comparing Figures 3c, 4c, and 5c, the epitopes that are considered to have a significant impact in the prediction model differ across observation periods. What is the clinical significance of the change in important epitopes over time during the tolerance acquisition? Is this merely a statistical error, or does it imply that the epitopes gradually shift to adjacent regions? It should be discussed in light of the positional relationship of the epitope in the allergen molecules

This observation could be interesting a priori, but it must be considered that our peptides have 20 AA with an offset of 3 AA, so two consecutive peptides share a 17AA core. It is therefore logical that several consecutive peptides share the recognition pattern and have the same weight in the prediction model. This would not reflect a temporal change, but rather the importance of that particular epitope.

Reviewer 3 Report

Comments and Suggestions for Authors

Thank you for the opportunity to review the manuscript titled, Prediction of Tolerance in Children With Cow's Milk Allergy Using Microarray Profiling, submitted as an article to Cells. In this manuscript, the authors aimed to “identify peptide biomarkers predictive of tolerance in a large Spanish population of children with [cow’s milk allergy] CMA” (copied verbatim from the abstract). This manuscript is generally well written, and I have only a few comments.

1.      Abstract: The authors note that CMA is the most common food allergy in infants. Yet, they do not provide the baseline age of participants (not patients). Are participants indeed infants, or are they older? This must be clarified.

2.      Abstract: This text contains many abbreviations, including some that are used without being introduced. Critically consider if all abbreviations are necessary, and be sure to introduce those which are retained in the text.

3.      Introduction: While the authors correctly note that most children will achieve tolerance, or possibly even resolution of their CMA, those who don’t often experience severe symptoms and lifelong persistence of disease. This must be noted in the first paragraph of the introduction.

4.      Introduction: The authors are encouraged to report that IgG4 remains controversial in allergic disease (e.g. PMID: 36389804)

5.      Results: The results are generally well done. However, I miss comments about the clinical history of these infants: history of reaction, family history of allergic disease, etc. This history would be interesting to overlay on the results.

6.      Minor comment: The authors refer to both the COALE and CoALE cohort. Which one is correct? Also, please define the abbreviation.

Author Response

Comments and Suggestions for Authors

Thank you for the opportunity to review the manuscript titled, Prediction of Tolerance in Children With Cow's Milk Allergy Using Microarray Profiling, submitted as an article to Cells. In this manuscript, the authors aimed to “identify peptide biomarkers predictive of tolerance in a large Spanish population of children with [cow’s milk allergy] CMA” (copied verbatim from the abstract). This manuscript is generally well written, and I have only a few comments.

Dear reviewer,

Thank you for your careful review of our paper and for the valuable comments, corrections, and suggestions that have helped us to improve the manuscript. Below, you will find the point-by-point response to your questions.

  1. Abstract: The authors note that CMA is the most common food allergy in infants. Yet, they do not provide the baseline age of participants (not patients). Are participants indeed infants, or are they older? This must be clarified.

Following your recommendation, we add a new supplementary table with the clinical and demographic information. (Table S1).

  1. Abstract: This text contains many abbreviations, including some that are used without being introduced. Critically consider if all abbreviations are necessary, and be sure to introduce those which are retained in the text.

New abbreviations and acronyms have been introduced. Page 12, line 404.

  1. Introduction: While the authors correctly note that most children will achieve tolerance, or possibly even resolution of their CMA, those who don’t often experience severe symptoms and lifelong persistence of disease. This must be noted in the first paragraph of the introduction.

Following your recommendation, we add the following sentence in the first paragraph of the Introduction:

“These patients in particular often experience severe symptoms and lifelong persistence of the disease.” Page 2, line 45.

  1. Introduction: The authors are encouraged to report that IgG4 remains controversial in allergic disease (e.g. PMID: 36389804)

Following your recommendation, we have added the following sentences and the reference in the Introduction:

“Although high levels of IgG4 antibodies have been associated with earlier development of tolerance, the role of IgG4 in allergic disease is controversial. It has been described that allergen-specific IgG4 plays a multifaceted role and may have a protective or pathogenic role depending on the allergens or exposure conditions [16].“ Page 2, line 59

  1. Results: The results are generally well done. However, I miss comments about the clinical history of these infants: history of reaction, family history of allergic disease, etc. This history would be interesting to overlay on the results.

Following your recommendation, we have added a new supplementary table with the clinical and demographic information (Table S1).

  1. Minor comment: The authors refer to both the COALE and CoALE cohort. Which one is correct? Also, please define the abbreviation.

CoALE is the acronym of the proyect: Cow milk ALEgy. We have corrected the acronym throughout the text.

Reviewer 4 Report

Comments and Suggestions for Authors

This study presents some interesting data, however some major improvements are necessary to properly display and discuss the data. General consistency and a clear storyline are lacking.

- The title of this study is more descriptive of the methods rather than the main outcomes, which is at the expense of the potential value of this manuscript. 

- In the introduction the assumption is made that CM is the first food to be introduced, please support this rather bold statement. Is this specific for Spain or Europe? 

- The design of the study is not sufficiently described. Referring to an unpublished manuscript does not provide enough information on the patient population. 

- Proper description of the patient population is necessary. During the manuscript it remains unclear whether all included patients have CMA at inclusion, the age range at inclusion become clear too late. What is meant with incident cases (line 61/62)?

- The subsection titles in the results section should be descriptive of the outcomes rather than the methods. 

- In line 130-133 explain why the percentage of tolerance is decreasing, while line 120/121 mentions all patients have CMA? Furthermore, in line 132/133 the age range is described in an unclear manner. 

- Table 1 should also contain total IgE and IgG4 levels. Why are the specific IgE and IgG4 caseins not subdivided in this table, but only total casein is shown?

- What is the value of an informative epitope? Please discuss in discussion why the informative epitopes may be relevant?

- Many patients develop tolerance between visit 3 and 4. Elaborate on this in your discussion how this could be explained? This is also the moment during which a change in predominant variable occurs... could this be related? If so, would this be related to tolerance development or is this a specific characteristic of sustained allergy? 

- The discussion describes two similar studies by Savilahti and Caubet. How do the outcomes of these studies support of contradict these presented data? This is now unclear. 

- Throughout the manuscript it is mentioned several times that there are no differences at baseline, however somehow this baseline has predictive value... but how? e.g. Lines 285/286. 

- Line 294/295 mentions that this study helps to identify patients who may benefit from immunotherapy, but explain which patient population is meant here. 

- It is mentioned that a large Spanish population is used. However including 118 patients, and this number significantly decreases during the study, cannot be really considered as a large population. 

Comments on the Quality of English Language

Use of English is okay, a check by a native English speaker is recommended to remove some errors. 

Author Response

Comments and Suggestions for Authors

This study presents some interesting data, however some major improvements are necessary to properly display and discuss the data. General consistency and a clear storyline are lacking.

Dear reviewer,

Thank you for your careful review of our paper and for the valuable comments, corrections, and suggestions that have helped us to improve the manuscript. Below, you will find the point-by-point response to your questions.

- The title of this study is more descriptive of the methods rather than the main outcomes, which is at the expense of the potential value of this manuscript.

Following your recommendation, we have changed the title as follows:

“Predicting tolerance to cow's milk allergy in children using IgE and IgG4 peptide binding profiles”

- In the introduction the assumption is made that CM is the first food to be introduced, please support this rather bold statement. Is this specific for Spain or Europe?

You are right, it refers mainly to Western societies. We modify that sentence accordingly and we have included a reference to support this:

“Cow's milk (CM) is the first food introduced after birth or breastfeeding in western societies, so it is the first food antigen with which one comes into contact.” Page 1, line 40.

- The design of the study is not sufficiently described. Referring to an unpublished manuscript does not provide enough information on the patient population.

To clarify this point, we have added a new supplementary figure showing the study design (Figure S1) and a new supplementary table with the clinical and demographic information of the patient population (Table S1).

- Proper description of the patient population is necessary. During the manuscript it remains unclear whether all included patients have CMA at inclusion, the age range at inclusion become clear too late. What is meant with incident cases (line 61/62)?

To clarify this point, a new supplementary table with the clinical and demographic information of the patient population (Table S1).

An incident case is a case that meets the inclusion criteria and has no exclusion criteria, and is included consecutively.

- The subsection titles in the results section should be descriptive of the outcomes rather than the methods.

Following your suggestion, we have changed the subtitle in the Methos as follows:

“3.2. IgE and IgG4 antibody binding to sequential epitopes revealed four informative epitopes recognized in all-time points” Page 4, line 156.

“3.3. Combining serological variables and peptide biomarkers at baseline can predict tolerability at later time points” Page 7, line 204.

- In line 130-133 explain why the percentage of tolerance is decreasing, while line 120/121 mentions all patients have CMA? Furthermore, in line 132/133 the age range is described in an unclear manner.

It is not that less and less patients were tolerant from visit to visit, but rather that, according to the protocol, tolerant patients finish the study and do not return for subsequent visits. At the end of the study only 13 patients remained allergic, while 76 were tolerant and 29 voluntarily withdrew from the study (lost patients). To clarify this point, we have added a new supplementary figure showing the study design and clearly indicating which patients are tolerant, allergic, withdrawing from the study, and the accumulated tolerant patients at each visit. (Figure S1).

To make this point clearer, we have added the following sentence:

“At the end of the study, 13 patients remained allergic, 76 were tolerant and 29 voluntarily withdrew from the study (lost patients) (Figure S1).” Page 4, line 150.

- Table 1 should also contain total IgE and IgG4 levels. Why are the specific IgE and IgG4 caseins not subdivided in this table, but only total casein is shown?

We have included the total IgE values in Table 1. IgG is not routinely measured and therefore we have no data on this.

- What is the value of an informative epitope? Please discuss in discussion why the informative epitopes may be relevant?

Following your suggestion, we have added the following paragraph to the Discussion to highlight the importance of informative epitopes:

“In the present work, we identified four key epitopes recognized by more than seventy-five percent of milk-allergic patients over time, suggesting a link to allergy persistence. In addition, we developed two predictive models of tolerance at 6 and 30 months, using the baseline IgE and IgG4 antibody-binding peptides in conjunction with standard serological variables. These biomarker peptides may be useful in the development of new diagnostic tests, peptide-based synthetic vaccines, or hypoallergenic milk formulas.” Page 9, line 269

We also modify the conclusion accordingly:

“In the present work, we identified four key epitopes recognized by more than seventy-five percent of milk-allergic patients over time, suggesting a link to allergy persistence. These epitopes may be good candidates for the development of peptide vaccines or hypoallergenic milk formulas. In addition, we developed two predictive models of tolerance at 6 and 30 months, using the baseline IgE and IgG4 anti-body-binding peptides in conjunction with standard serological variables.” Page 11, line 359.

- Many patients develop tolerance between visit 3 and 4. Elaborate on this in your discussion how this could be explained? This is also the moment during which a change in predominant variable occurs. could this be related? If so, would this be related to tolerance development or is this a specific characteristic of sustained allergy?

Our results are in agreement with previous studies which establish that CMA is usually a transient disease, with most children outgrowing their allergy by 3 years of age and only a small percentage persisting beyond 4 years. It is therefore expected that most of the changes will occur at these times, and that there will be fewer changes at later times, probably associated with persistent allergy. We therefore believe that these changes are related to the natural evolution of the disease and to the spontaneous development of tolerance.

To better clarify this point we added the following sentences to the Discussion:

“This fact is related to natural tolerance development, so in our study most children outgrow their allergy by 3 years of age, and only a small percentage persisting beyond 4 years [24].” Page 9, line 265.

- The discussion describes two similar studies by Savilahti and Caubet. How do the outcomes of these studies support of contradict these presented data? This is now unclear.

As we mentioned in the discussion, these two studies have a very different design to our study, and therefore, their results are very difficult to compare. What they do agree on is that all three were able to associate the recognition of some epitopes with the development of tolerance. We believe that we cannot go beyond what is discussed on page 10, line 298.

- Throughout the manuscript it is mentioned several times that there are no differences at baseline, however somehow this baseline has predictive value... but how? e.g. Lines 285/286.

This statement refers to the global response of peptide recognition, analyzed as global intensity and diversity for each patient, which is commonly used in microarray studies of allergenic peptides to describe patient behavior and its relationship to study outcome. However, this does not mean that individual peptides show differences and, therefore, have predictive value.

We have found that the sentence on page 9 line 276 does not clearly reflect this idea and have amended it accordingly:

“In agreement with these studies, we also observed no differences in IgE and IgG4 intensity and diversity of peptide recognition between patients at the time of diagnosis.” Page 10, line 304.

- Line 294/295 mentions that this study helps to identify patients who may benefit from immunotherapy, but explain which patient population is meant here.

In the present work, we have proposed two models to predict at the time of diagnosis which patients will tolerate milk at a later stage, so that patients at high risk of not tolerate could benefit from early intervention with immunotherapy without having to wait for persistent allergy to develop. However, as we mentioned, these results should be treated with caution as further research is needed to validate the use of these peptide biomarkers.

- It is mentioned that a large Spanish population is used. However including 118 patients, and this number significantly decreases during the study, cannot be really considered as a large population.

You are right, we have removed the word "large" from this sentence, page 1, line 22.

Round 2

Reviewer 1 Report

Comments and Suggestions for Authors

All in all, the changes made improve to reader's understanding of the methodology and findings of this study.

A few minor remarks:

Figure S1 – adds important information and helps understanding study design. I think it should be included in the main manuscript and NOT as supplementary material. Table 1 can be moved to the supplementary material.

I also suggest putting all font in Black, since the white font is hard to read over the grey background

Discussion-

The sentence in line 269- " In the present work, we identified four key epitopes recognized by more than seventy-five percent of milk-allergic patients …." Please mention the specific key epitopes

Line 90- I would remove the mentioning of the quality of life questionnaires since they are of no significance to this study.

Line 256- The opening of the discussion- please delete the "bb" that appear at the start of the sentence

Lines 365-370- there is repetition in the sentences, please correct.

Conclusion- I still think that a "bottom line" is missing. I would specify the epitopes found and even choose one or two that are most relevant for future studies.

Author Response

Dear reviewer, thank you for your valuable comments and suggestions for improving the manuscript.

A few minor remarks:

Figure S1 – adds important information and helps understanding study design. I think it should be included in the main manuscript and NOT as supplementary material. Table 1 can be moved to the supplementary material.

Following your recommendation, we move Figure S1 to the main manuscript.

Following the suggestion of reviewer 2, we have also combined Table 1 and S1 into a single one, Table 1.

I also suggest putting all font in Black, since the white font is hard to read over the grey background

Following your suggestion, we have removed the grey boxes and replaced them with white text.

Discussion-

The sentence in line 269- " In the present work, we identified four key epitopes recognized by more than seventy-five percent of milk-allergic patients …." Please mention the specific key epitopes

Following your suggestion, we have added the specific key epitopes.

“In the present work, we identified four key epitopes (αs1-cas 3, αs1-cas 16-18, κ-cas 29-31, and β-lac 37-39) recognized by more than seventy-five percent of milk-allergic patients over time, suggesting a link to allergy persistence.” Page 10, line 279.

 Line 90- I would remove the mentioning of the quality of life questionnaires since they are of no significance to this study.

Following your recommendation, we have removed the FAQLQ-PF Quality of Life Questionnaire. Page 2, line 91.

Line 256- The opening of the discussion- please delete the "bb" that appear at the start of the sentence

We have corrected that in the text.

Lines 365-370- there is repetition in the sentences, please correct.

We have corrected this in the text as follows:

“If the linear B epitopes identified in our study are validated as indicators of CM tolerance or persistence, their determination will allow us to have personalized and precise therapeutic management from the onset of CMA in the future.” Page 12, line 375.

Conclusion- I still think that a "bottom line" is missing. I would specify the epitopes found and even choose one or two that are most relevant for future studies.

Following your suggestion, we have added the specific key epitopes:

“In the present work, we identified four key epitopes (αs1-cas 3, αs1-cas 16-18, κ-cas 29-31, and β-lac 37-39) recognized by more than seventy-five percent of milk-allergic patients over time, suggesting a link to allergy persistence.” Page 12, line 380.

Reviewer 2 Report

Comments and Suggestions for Authors

Revisions and additions have clarified the conclusions of this study. 

Since there is some overlap between Supplementary Table 1 and Table 1, it may be better to combine them into one table. Alternatively, the age and gender information in Supplementary Table 1 could be deleted.

Author Response

Revisor 2

Dear reviewer, thank you for your valuable comments and suggestions for improving the manuscript.

Comments and Suggestions for Authors

Revisions and additions have clarified the conclusions of this study. 

Since there is some overlap between Supplementary Table 1 and Table 1, it may be better to combine them into one table. Alternatively, the age and gender information in Supplementary Table 1 could be deleted.

Following your recommendation, we combine Table 1 and S1 into Table 1.

Reviewer 4 Report

Comments and Suggestions for Authors

The revisions have clearly improved the manuscript. 

Since the context and study design have become more clear, data can be interpreted in the context. 

Author Response

Dear reviewer, thank you for your valuable comments and suggestions for improving the manuscript.